

# Montreal Cognitive Assessment (MoCA) performance in Huntington's disease patients correlates with cortical and caudate atrophy

Gabriel Ramirez-Garcia[1], Victor Galvez[2], Rosalinda Diaz[1], Aurelio Campos-Romo[3] and Juan Fernandez-Ruiz[1]

[1] Departamento de Fisiología, Universidad Nacional Autónoma de Mexico, Ciudad de Mexico, Mexico
[2] Escuela de Psicología, Universidad Panamericana, Ciudad de Mexico, Mexico
[3] Facultad de Medicina, Unidad Periférica de Neurociencias, Universidad Nacional Autónoma de México/Instituto Nacional de Neurologia y Neurocirugia, Ciudad de Mexico, Mexico

Corresponding author
Juan Fernandez-Ruiz, jfr@unam.mx

## ABSTRACT

Huntington's Disease (HD) is an autosomal neurodegenerative disease characterized by motor, cognitive, and psychiatric symptoms. Cognitive impairment develops gradually in HD patients, progressing later into a severe cognitive dysfunction. The Montreal Cognitive Assessment (MoCA) is a brief screening test commonly employed to detect mild cognitive impairment, which has also been useful to assess cognitive decline in HD patients. However, the relationship between MoCA performance and brain structural integrity in HD patients remains unclear. Therefore, to explore this relationship we analyzed if cortical thinning and subcortical nuclei volume differences correlated with HD patients' MoCA performance. Twenty-two HD patients and twenty-two healthy subjects participated in this study. T1-weighted images were acquired to analyze cortical thickness and subcortical nuclei volumes. Group comparison analysis showed a significantly lower score in the MoCA global performance of HD patients. Also, the MoCA total score correlated with cortical thinning of fronto-parietal and temporo-occipital cortices, as well as with bilateral caudate volume differences in HD patients. These results provide new insights into the effectiveness of using the MoCA test to detect cognitive impairment and the brain atrophy pattern associated with the cognitive status of prodromal/early HD patients.

## INTRODUCTION

Huntington's Disease (HD) is an autosomal dominant inherited neurodegenerative disease (*Huntington's Disease Collaborative Research Group, 1993*). The hallmark of the HD neuropathology is caudate neurodegeneration; although several cortical regions are also affected (*Rosas et al., 2008*; *Tabrizi et al., 2009*; *Aylward, 2014*). HD clinical manifestations include motor, cognitive, and psychiatric deficits. Even though HD is classically considered as a motor disorder, the cognitive decline may occur even before the clinical manifestations

of the motor symptoms start (*Paulsen, 2011*). Although it has been suggested that cognitive deficits are mainly linked to striatal degeneration (*Kassubek et al., 2004*; *Peinemann et al., 2005*; *Aylward et al., 2013*), the extensive atrophy of the cortical mantle also contributes considerably to the progression of these alterations (*Johnson et al., 2015*).

Cognitive assessment in HD patients has been performed using several tests, including the Cognitive Functioning evaluation of the Unified Huntington's Disease Rating Scale (UHDRS) (*Mestre et al., 2016*). However, the analysis of the possible brain structures contributing to the cognitive impairment in HD has only been based on correlation analyses with the UHDRS cognitive component, namely, no other conventional cognitive assessment instrument has been associated with the structural brain atrophy in HD. One of the most used cognitive screening tests in research settings and clinical practice is the Montreal Cognitive Assessment (MoCA). This screening test shows a high sensitivity to detect mild cognitive impairment (MCI) in the evaluated participant (*Nasreddine et al., 2005*). Besides, the MoCA global performance can be used to detect, follow up, and predict the progression of cognitive impairment in a number of clinical populations (*Julayanont et al., 2014*; *Sivakumar et al., 2014*; *Nijsse et al., 2017*; *Hendershott et al., 2017*). Likewise, the MoCA total score has also shown a relationship with MRI functional markers associated with cognitive impairment (*Cheewakriengkrai et al., 2014*; *Porges et al., 2017*) and with atrophy of cortical and subcortical nuclei (*Ritter et al., 2017*; *Ogawa et al., 2019*). The aforementioned evidence suggests that the MoCA test is useful not only to evaluate cognitive performance but also to explore its relationship with functional and structural brain deterioration.

In HD patients the MoCA test has been evaluated to determine its usefulness and sensitivity to detect the presence of potential cognitive impairment. Some reports have demonstrated that the MoCA test presents high sensitivity and specificity to detect cognitive impairment, even suggesting that it presents higher sensitivity than other brief screening instruments (*Videnovic et al., 2010*; *Mickes et al., 2010*; *Ringkøbing et al., 2020*). Consequently, the MoCA test is a suitable monitoring and screening tool for assessing cognitive dysfunction in HD patients (*Videnovic et al., 2010*; *Mickes et al., 2010*; *Bezdicek et al., 2013*; *Gluhm et al., 2013*; *Ringkøbing et al., 2020*) across a wide range of disease severity stages (Mild, Moderate, and Severe) (*Gluhm et al., 2013*). However, there is no current evidence of the association of MoCA global performance with the pattern of brain structural atrophy exhibited by HD patients. Therefore, the present study aimed to determine the relationship between the MoCA global performance and the cortical and subcortical gray matter deterioration in prodromal/early HD patients.

## METHODS

### Participants

Twenty-two HD patients and twenty-two healthy controls matched for age, sex, and years of education participated in this study (Table 1A). All HD patients had a positive molecular diagnosis and were invited to participate in this study at the Instituto Nacional de Neurología y Neurocirugía "Manuel Velasco Suárez". Healthy volunteers self-reported no history of neurological or psychiatric disorders and were recruited at the same period as

**Table 1 Sample and clinical characteristics.**

| A. Sample characteristics | Healthy controls | | | Huntington's disease patients | | | Statistical significance | | | Effect size Cohen's d or r |
|---|---|---|---|---|---|---|---|---|---|---|
| | Mean | SD | Range | Mean | SD | Range | t-value | W-value | p-value | |
| Male:Female ratio | 9:13 | —— | —— | 9:13 | —— | —— | —— | —— | —— | —— |
| Sample size (n) | 22 | —— | —— | 22 | —— | —— | —— | —— | —— | —— |
| Age (years) | 45.47 | 12.24 | 40.08 | 46.11 | 12.11 | 41.08 | −0.17 | —— | 0.863[NS] | −0.05 S |
| Handedness (R/L/B) | 22/0/0 | —— | —— | 21/1/0 | —— | —— | —— | —— | —— | —— |
| Education (years) | 16.13 | 2.85 | 10 | 14.13 | 3.21 | 11 | —— | 305.5 | 0.133[NS] | 0.22 S |
| Disease Burden | —— | —— | —— | 389.96 | 100.58 | 379.19 | —— | —— | —— | —— |
| CAG repeat length | —— | —— | —— | 44.59 | 3.82 | 14 | —— | —— | —— | —— |
| ICV (cm$^3$) | 1384.17 | 145.88 | 566.42 | 1276.49 | 125.91 | 370.03 | —— | 352 | **0.009**[**] | 0.38 M |
| **B. Clinical measures** | | | | | | | | | | |
| CES-D | 8.89 | 4.90 | 18 | 10.81 | 6.91 | 25 | −1.01 | —— | 0.310[NS] | −0.31 S |
| UHDRS-TMS | —— | —— | —— | 17.18 | 13.63 | 58 | —— | —— | —— | —— |
| TFC | —— | —— | —— | 11.90 | 1.82 | 5 | —— | —— | —— | —— |

Note:
The significant differences are highlighted in bold (**$p < 0.01$). CES-D, Center for Epidemiologic Studies Depression Scale; TFC, Total Functional, Capacity scale; UHDRS-TMS,Unified Huntington's Disease Rating Scale-Total Motor Score. ICV, Intracranial volume. SD, Standard Deviation. R, Right; L, Left; B, Both. Effect size: S, Small; M, Medium. NS, Not Significant.

the HD-subject group. All the procedures were performed according to the Declaration of Helsinki and approved by the health and ethics committees of the Instituto Nacional de Neurología y Neurocirugía "Manuel Velasco Suarez" (N° DIC/419/14 and N° 41/14) and Universidad Nacional Autónoma de Mexico (N° 090/2015). All participants provided written informed consent before participating in the study.

## Clinical assessment

For all HD patients, the functional and motor status were evaluated using the Total Functional Capacity (TFC) scale (*Shoulson & Fahn, 1979*) and the Total Motor Score (TMS), respectively; both instruments from the UHDRS (*Huntington Study Group, 1996*). These clinical outcomes were used to measure the stage and severity of HD. Accordingly, the staging of patients follows the TFC scoring where scores from 11–13 represent stage I (least severe); 7–10, stage II; 3–6, stage III; 1–2, stage IV; and score of 0 is stage V (most severe) (*Paulsen et al., 2010*). The authors received permission to use this instrument from the copyright holders (Huntington Study Group). The Spanish version of the Center for Epidemiologic Studies Depression Scale (CES-D) (*Radloff, 1977*; *Soler et al., 1997*; *Unschuld et al., 2012*) was used as an indicator of a depressed mood and was administered to all participants. The sum of all items ranges from 0 to 60 and scores ≥16 are considered as an indicator of depressive symptoms. This scale is free to use without permission.

## MoCA test

MoCA test was used to evaluate cognitive status in both groups. This cognitive screening test consists of twelve individual tasks grouped into eight individual sections: visuospatial/ executive, naming, memory, attention, language, abstraction, delay recall, and orientation.

The MoCA total score (30 pts) reflects the global cognitive performance, and it is calculated by summing the individual sections scores plus an additional educational level correction (*Nasreddine et al., 2005*; *Aguilar-Navarro et al., 2018*). In HD patients, the recommended cut-off score ≥26 pts (*Bezdicek et al., 2013*; *Ringkøbing et al., 2020*; *Rosca & Simu, 2020*) was further validated and used for screening suspected MCI. The authors received permission to use this instrument from the copyright holders (https://www.mocatest.org/).

## Image acquisition

The high-resolution T1-weighted anatomical images were obtained with a Fast Field-Echo sequence with the following parameters: TR/TE: 8/3.7 ms; FOV: $256 \times 256$ mm$^2$; flip angle: 8°, acquisition, and reconstruction matrix: $256 \times 256$; isometric resolution: $1 \times 1 \times 1$ mm$^3$. All brain images were acquired using a 3T Achieva MRI scanner (Phillips medical systems, Eindhoven, The Netherlands) at Instituto Nacional de Psiquiatría "Ramón de la Fuente Muñiz" in Mexico City. The image preprocessing included: MNI orientation, denoising, and intensity inhomogeneity correction (*Manjón et al., 2010*; *Avants et al., 2011*).

## Subcortical nuclei volumes quantification

Volumes of subcortical nuclei were extracted using a patch-based method by an automated volume system implemented in VolBrain online web interface (http://volbrain.upv.es) (*Manjón & Coupé, 2016*). The volume was obtained for fourteen (bilateral) subcortical regions of interest (ROI) including caudate, putamen, thalamus, globus pallidus, hippocampus, amygdala, and nucleus accumbens. For each individual, subcortical nuclei volumes were calculated as a percentage of their respective intracranial volume (ICV). Brain masks for ICV calculation were inspected visually to determine possible intracranial cavity extraction errors. No errors were found.

## Cortical reconstruction and cortical thickness determination

Cortical reconstruction was performed using FreeSurfer image analysis suite (http://surfer.nmr.mgh.harvard.edu/) version 7.2. The fully automated procedure includes the following steps: non-uniform intensity normalization, correction of intensity variations due to magnetic field inhomogeneity, skull stripping, segmentation, separation of left and right hemispheres as well as cortical from subcortical structures, triangular tessellation of the grey matter-white matter (GM-WM) boundary, topology correction, deformable data processing, and surface inflation registration to a spherical atlas, parcellation of the cerebral cortex, and creation of a variety of surface-based data. Finally, cortical thickness was computed by using the distance between the gray/white surface and the pial surface (*Fischl et al., 1999*; *Fischl, Sereno & Dale, 1999*; *Fischl & Dale, 2000*; *Fischl, Liu & Dale, 2001*). The quality control of the cortical reconstruction involved a visual inspection of each image to detect potential topological defects. For all subjects' images, pial surface or WM segmentation errors were found; therefore, each error was corrected manually, and cortical reconstruction was rerun; finally, the new WM and pial surfaces were inspected again to avoid further errors. The correlations between MoCA total score and cortical thickness were performed using disease burden, ICV, and years of education as nuisance

factors for the HD patients, and ICV and years of education for the healthy controls. All analyses were performed vertex-wise in the whole brain cortex by the General Linear Model within Qdec to model the data. Correction for multiple comparisons for both analyses was performed by permutation testing using Monte Carlo Simulation with a smoothing of 15-mm full width at half height Gaussian kernel setting a significance level of $p < 0.05$. Anatomical labels were obtained from the Desikan-Killiany cortical atlas.

## Statistical analysis

We evaluated the data normality with the Shapiro-Wilk test ($p < 0.05$) using the standardized residuals after a linear regression for group comparison analysis and using the raw data for correlation analysis. In addition, normality distribution was analyzed visually by a quantile-quantile plot of each variable *vs* a normal distribution. After these analyses, the appropriate statistical tests to be performed were determined. If the *p*-value was less than 0.05 (normality distribution is not assumed) a non-parametric test was implemented, but if the *p*-value was higher than 0.05 (normality distribution is assumed) a parametric analysis was implemented. Comparison between healthy controls and HD patients for age, years of education, ICV, CES-D scores, and MoCA total score were analyzed by two-tailed t-test or Mann-Whitney U-test as appropriate, setting a significance level of $p < 0.05$. It is important to mention that for the CES-D scores for the control group the sample size was 19 subjects because data from three subjects was unavailable from the hospital; for the rest of the comparison the sample size for the control and HD groups was 22 subjects each. For HD patients, partial correlations between MoCA total score and CAG repeat length, CES-D score, UHDRS-TMS score, and TFC score were computed by Spearman's rank correlation rho, and the correlations with subcortical nuclei volumes were computed by Pearson's correlation. All partial correlations included the disease burden and years of education as nuisance factors, except the MoCA correlation with CAG repeat length, which only included years of education as a nuisance factor. For healthy subjects, partial correlations of MoCA total score with subcortical nuclei volumes were computed by Pearson's correlation including the years of education and age as nuisance factors. For all comparisons, the effect size was computed by Cohen's *d* for parametric test ($d = 0.2–0.50$ Small; $d = 0.50–0.80$ Medium; larger than $d = 0.80$ Large) and *r* for nonparametric tests ($r = 0.10–0.30$ Small; $r = 0.30–0.50$ Medium; larger than $r = 0.50$ Large). Disease burden was computed by the formula: age (years) × (CAG repeat length − 35.5) (*Penney et al., 1997*). The analysis of the area under the curve (AUC) from the receiver operating characteristic (ROC) was performed for the MoCA screening test to corroborate the cut-off score in our study, setting a significance level of $p < 0.05$ and a confidence interval (CI) of 95%. All statistical analyses were performed using R 3.6.0 and RStudio Version 1.1.463.

## RESULTS

### Demographic and clinical characteristics

There were no statistical differences in age or years of education between healthy controls and HD patients (Table 1A), but significant differences were found in ICV ($p = 0.009$) with

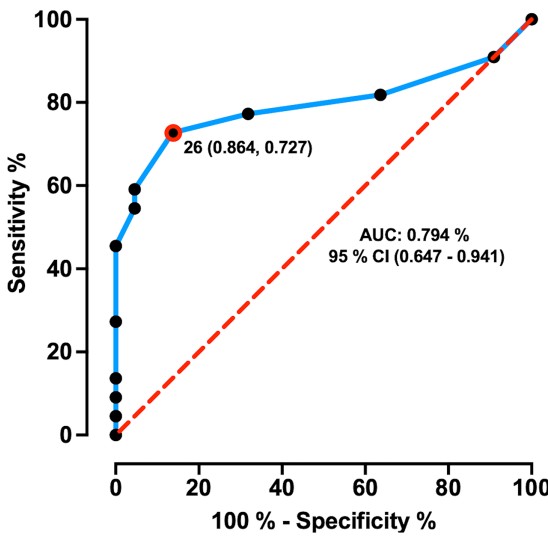

**Figure 1 ROC curve of MoCA total score in prodromal/early HD patients.** ROC curve distinguishing between healthy controls and HD patients. ROC curve plotted the true positive rate (Sensitivity %) in the function of the false positive rate (100%-Specificity %). The dashed red line represents the area under the curve of 0.5 and the "J" point (red point) stands for the Youden index, *i.e.*, the point with the best combination of Sensitivity and Specificity.

a medium effect size (Cohen's $r$ = 0.38), indicating a moderated difference between group medians. There was no statistical difference in CES-D score between healthy controls and HD patients. The functional and motor status of HD patients indicated an initial symptomatic stage. Most of the patients were in clinical stage I (82%; 11–13 pts), and only four of them were in stage II (18%; 7–10 pts) according to their TFC performance. In addition, HD patients' motor status (UHDRS-TMS) coincided with the disease stage I (Mean ± SD = 12.38 ± 7.86) and stage II (38.75 ± 13.86), showing a mild severity of motor signs (Table 1B). Based on these scores, the HD cohort was considered prodromal/early HD patients (Prodromal $n$ = 16; Early $n$ = 6).

## MoCA cut-off determination and MoCA global performance comparison

The AUC ± SD of MoCA total score was 0.794 ± 0.07 with 95% CI of [0.647–0.941], and sensitivity and specificity of 72.7% and 86.4%, respectively. MoCA test showed an AUC significantly greater than 0.5 ($p$ = 0.000), which demonstrates its discriminative ability between HD patients and healthy controls. The maximum likelihood of Youden's index J was 0.59 and showed the optimum cut-off point was < 26 pts coinciding with its suggested original cut-off score (Fig. 1). Comparison for MoCA total score between healthy control (Mean ± SD = 27.90 ± 1.41; range = 6) and HD patients (Mean ± SD = 24.18 ± 4.17; range = 16) showed significant differences ($W$ = 384.5; $p$ = 0.000) with a large effect size (Cohen's $r$ = 0.50; Fig. 2A), indicating a large difference between group medians. Thirteen HD patients (59%) presented an impaired cognitive performance with scores ranging between 14 to 25 pts, and nine patients (41%) had no impaired cognitive performance with MoCA scores ≥ 26 pts.

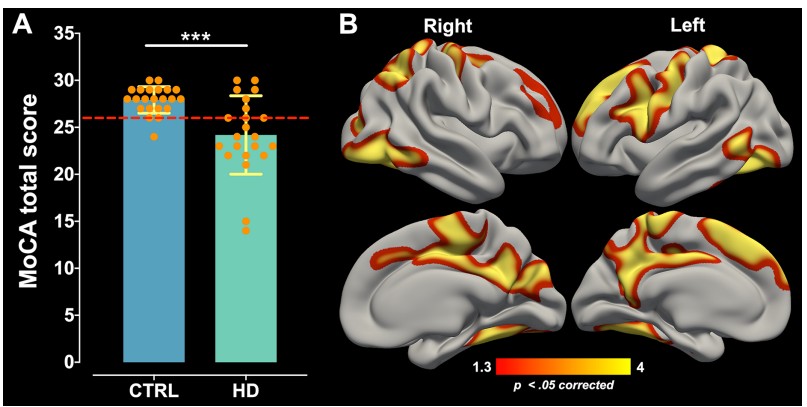

**Figure 2 MoCA global performance difference and its correlation with cortical thinning.** (A) MoCA total score comparison between the control group (CTRL) and HD patients. Data are shown as mean ± SD (***$p$ = 0.000). The dashed red line represents the MoCA cut-off. All subjects are represented as individual data points. (B) The positive correlation between MoCA global performance and cortical thinning across the whole cortical mantle of HD patients. Significant cluster maps are projected onto an average cortical surface. The warm color scale represents the significance values corrected by Monte Carlo simulation ($p$ < 0.05).

**Table 2 MoCA correlation with clinical measures of HD patients.**

| MoCA test correlations | $\rho$-coefficient | $p$-value |
|---|---|---|
| CAG repeat length | 0.35 | 0.110[NS] |
| CES-D | 0.11 | 0.620[NS] |
| TFC | 0.73 | **0.000***** |
| UHDRS-TMS | −0.78 | **0.000***** |

**Note:**
The significant differences are highlighted in bold (***$p$ < 0.001). NS, Not Significant.

## Correlation of MoCA global performance with CAG repeat length and clinical measures

MoCA total score showed a significant strong positive correlation ($\rho$ = 0.73, $p$ = 0.000) with functional status (TFC) and a significant strong negative correlation ($\rho$ = −0.78, $p$ = 0.000) with motor performance (UHDRS-TMS) of HD patients, which indicates that there is a strong linear relationship between the variables. There was no significant correlation with CAG repeat length, or CES-D score (Table 2).

## Correlation of cortical thinning and subcortical nuclei volumes and MoCA global performance

Correlation analysis performed between cortical thickness and MoCA total score of the HD patients revealed positive significant associations in clusters located bilaterally in the superior frontal, rostral middle frontal, precentral, superior parietal, lateral occipital, middle temporal, inferior temporal, isthmus cingulate, paracentral, posterior cingulate, precuneus, and fusiform cortical areas; significant correlations were also found in clusters located only in the left hemisphere including caudal middle frontal, postcentral, pars opercularis, and banks of superior temporal sulcus cortical areas; and significant

**Table 3 MoCA global performance and subcortical nuclei volume correlation for HD patients.**

| MoCA test correlations | r-coefficient | p-uncorrected | p-value (Bonferroni correction) |
|---|---|---|---|
| Caudate R | 0.641 | 0.002 | **0.032**[*] |
| Caudate L | 0.647 | 0.002 | **0.028**[*] |
| Putamen R | 0.561 | 0.010 | 0.140[NS] |
| Putamen L | 0.602 | 0.004 | 0.068[NS] |
| Thalamus R | 0.399 | 0.080 | 1[NS] |
| Thalamus L | 0.367 | 0.110 | 1[NS] |
| Globus Pallidus R | 0.580 | 0.007 | 0.102[NS] |
| Globus Pallidus L | 0.534 | 0.015 | 0.212[NS] |
| Hippocampus R | 0.518 | 0.019 | 0.266[NS] |
| Hippocampus L | 0.494 | 0.026 | 0.370[NS] |
| Amygdala R | 0.600 | 0.005 | 0.071[NS] |
| Amygdala L | 0.692 | 0.000 | **0.010**[*] |
| Accumbens R | 0.465 | 0.038 | 0.540[NS] |
| Accumbens L | 0.513 | 0.020 | 0.288[NS] |

Note:
The significant differences are highlighted in bold ([*]$p < 0.05$). NS, Not Significant; R, Right; L, Left.

correlations were found for the right hemisphere in the cuneus, pericalcarine, and inferior parietal cortical areas (Fig. 2B; for a more detailed anatomical description see Supplemental Information–Fig. S1). The peak max of each significant clusters was located in precuneus, lateral occipital, inferior parietal, and rostral middle frontal cortical regions for the right hemisphere and superior frontal, precuneus, and fusiform cortical regions for the left hemisphere (see Supplemental Information– Tables S1 and S2). Partial correlations between subcortical volumes of 14 ROIs with the MoCA total score showed a significant strong positive association for the bilateral caudate (right: $r = 0.641$, $p = 0.032$, and left: $r = 0.647$, $p = 0.028$) and left amygdala ($r = 0.692$, $p = 0.010$), which indicates that there is a high linear relationship between these variables (Table 3). Partial correlations analysis performed between MoCA total score, and cortical thickness and subcortical nuclei volumes of healthy controls did not show any significant association (see Supplemental Information–Table S3).

## DISCUSSION

In this study, we tested the hypothesis that MoCA global performance correlates with a specific brain atrophy pattern in prodromal/early HD. Our analyses identified that the MoCA total score correlates with bilateral caudate and left amygdala volume differences and with extensive cortical thinning of areas that belong to the temporo-occipital, and parietal and superior frontal cortices.

MoCA has demonstrated adequate sensitivity to detect cognitive impairment in HD patients (*Mickes et al., 2010*; *Bezdicek et al., 2013*; *Gluhm et al., 2013*). Here, we corroborate the ability of MoCA to distinguish between healthy controls and prodromal/early HD patients according to their cognitive performance. Besides, we determine the appropriate

MoCA cut-off score (< 26 pts) for this HD cohort, which coincided with the original cut-off point suggested previously to HD (*Nasreddine et al., 2005*; *Bezdicek et al., 2013*; *Ringkøbing et al., 2020*).

The analysis of the prodromal/early HD patients' MoCA total score showed a significant difference with respect to the control group. This result support previous reports showing that symptomatic as well as prodromal HD patients exhibit cognitive decline assessed using brief screenings or comprehensive neuropsychological tests (*Rosas et al., 2008*; *Duff et al., 2010*; *Tabrizi et al., 2013*; *Paulsen, Smith & Long, 2013*; *You et al., 2014*). It should be noted that the MoCA test composition encompasses psychometric properties to detect cognitive impairment (*Vogel et al., 2015*); therefore, this test is particularly useful for HD because the patients typically develop notably heterogeneous cognitive deficits with different degrees of impairment that weigh the presence of the global cognitive decline.

MoCA global performance correlations analysis demonstrated extensive cortical areas associated with cognitive performance in HD patients. In this sense, our findings are consistent with previous reports in HD that identified specific cognitive deficits –attention, working memory, and executive functions– correlating with cortical changes including cortices of the superior fronto-parietal motor circuits, frontal cognitive control centers, temporal auditory and semantic processing hubs, and occipital visual centers (*Rosas et al., 2005*, *2008*; *Harrington et al., 2014*; *Coppen et al., 2018*; *Martinez-Horta et al., 2020*). It is important to mention that the MoCA test involves the assessment of several cognitive functions associated with different neurological substrates; therefore, it is understandable that the atrophy of discreet areas within the whole cerebral cortex contributed to the deterioration of MoCA global performance of the HD patients.

It is well known that striatal atrophy is the neuropathological hallmark of HD (*Vonsattel et al., 1985*; *Aylward et al., 2000*) being the caudate volume differences considered as a biomarker of disease progression (*Aylward, 2014*). Since the striatum is strongly associated with cognitive functioning (*Graff-Radford et al., 2017*), it was expected that the caudate volume differences correlated strongly with MoCA total score. Therefore, our results strengthen the previous evidence showing that caudate atrophy is associated with cognitive impairment evaluated by different neuropsychological tests (*Peinemann et al., 2005*; *Rosas et al., 2008*; *Aylward et al., 2013*; *You et al., 2014*; *Kim et al., 2017*). In this respect, it would be sensible to suggest that the influence of striatal atrophy on cognitive impairment is driven first by the progressive death of striatal spiny neurons; and secondly by the disconnection of this nucleus with the frontal cortex as a result of axonal neurodegeneration (*Rosas et al., 2010*, *2018*; *Poudel et al., 2015*) affecting cortico-striatal tracts highly connected to regions within the motor and associative networks (*Rosas et al., 2010*).

Conversely, the MoCA scores also correlated with the volume differences of the left amygdala, which has been of interest because of the evidence that the amygdala is affected in presymptomatic and symptomatic HD stages (*Douaud et al., 2006*; *Ahveninen et al., 2018*; *Tang et al., 2018*). Although the amygdala has been strongly associated with psychiatric symptoms (*Adolphs, 2000*; *Fine & Blair, 2000*), recently, it has been shown that smaller amygdala volumes are associated with worse visuomotor skills, slower processing speed, and emotional recognition (*Kipps et al., 2007*; *Ahveninen et al., 2018*). It is

important to note that in the healthy control group, we did not identify a significant correlation between MoCA global performance and cortical and subcortical gray matter differences; therefore, we suggest that the cognitive-atrophy associations found for HD patients are specific to the neuropathology of disease.

The MoCA total score also showed significant strong correlations with the clinical measures TFC and UHDRS-TMS; this is in line with previous reports showing that cognitive impairment is associated with the decline of functional and motor performance in HD patients (*Hamilton, 2003*; *Nehl & Paulsen, 2004*; *Jacobs et al., 2016*). In addition, the degree of cognitive impairment does increase proportionally with the HD stage progression determined by functional and motor status, which is reflecting a common behavioral phenotype of HD (*Toh et al., 2014*; *Jacobs et al., 2016*). Furthermore, it has been recently suggested that cognitive and motor symptoms share a common neurobiological basis (*Garcia-Gorro et al., 2019*), though these alterations may be affected at different levels and progress distinctively. Finally, even though CAG repeat length drives age of onset and severity of disease, and it has been correlated with volume differences of the striatum and motor cortex (*Rosas et al., 2001*; *Langbehn et al., 2019*) as well as with cognitive and motor deficit progression (*Rosenblatt et al., 2006*, *2012*), here we did not find a significant correlation with MoCA global performance. Further studies would be necessary to identify if MoCA decline presents a CAG repeat length–dependent trajectory.

To the best of our knowledge, this is the first study that evaluates the correlation of the MoCA global performance with the cortical brain atrophy of a neurodegenerative disease such as HD. However, our study had some limitations to consider. (1) It would be ideal to reproduce the correlation analysis with a larger sample size to corroborate the consistency of the significant clusters of cortical atrophy patterns and subcortical volume correlations. (2) For the future development of the project, it will be relevant to complement the clinical diagnosis of HD not only with the TFC and UHDRS-TMS performance but also with the Diagnostic Confidence Level (DCL), which is the standard measure used for clinical diagnosis in at-risk individuals and is based solely on the motor evaluation. (3) All HD patients were in a prodromal or early disease stage; therefore, we recommend implementing this analysis in presymptomatic, prodromal, and moderate manifest HD patients to clarify the atrophy-progressive pattern among different clinical disease stages, always considering the potential motion artifacts produced by a hyperkinetic condition present at manifest stages. (4) An extensive neuropsychiatric evaluation was not included. Therefore, we recommend carrying out a neuropsychiatric evaluation during the cognitive assessment session to incorporate it as a possible covariate in further analyses. (5) We also did not consider medication HD-related on cognitive analysis because most of the patients were not under specific treatment with a controlled and stable dosage given the prodromal/early disease stage (see Supplemental Information–Table S4).

## CONCLUSION

In summary, the MoCA test captures not only cognitive status but also structural brain atrophy associated with the HD cognitive impairment; particularly, the atrophy of the

caudate and the superior-lateral frontal areas as well as posterior regions such as temporoparietal and lateral occipital areas. This finding may result in a more detailed understanding of the neural bases of the specific cognitive deficits of HD patients.

Overall, the MoCA test could be used as the first approach in clinical practice given its ability for screening the cognitive impairment of HD patients; then, cognitive impairment detected may be addressed with other comprehensive cognitive batteries for further characterization. Moreover, the MoCA test is a suitable screening tool due to its relationship with the brain atrophy pattern exhibited by prodromal/early HD patients, which may help to establish adequate neuropsychological rehabilitation programs or to introduce individualized disease-modifying treatment plans according to the cognitive deficits detected and the stage of the disease.

### Funding

This study was supported by CONACYT–Mexico grant No. A1-S-10669 and PAPIIT-UNAM grant No. IN220019 to Juan Fernandez-Ruiz. CONACYT–Mexico Ph.D. fellowship No. 369794 and grant Fondo semilla-2019 No. CIP-PI-029-2018-2 from FCS-Universidad Panamericana given to Victor Galvez (CVU: 421958). Ramirez-Garcia received a doctoral (Programa de Doctorado en Ciencias Biomédicas, Universidad Nacional Autónoma de Mexico) fellowship 574022/403010 (CVU: 660496) and postdoctoral fellowship No. CB-2017-2018-A1-S-10669-M-5248 by the Fondo Sectorial de Investigación para la Educación Básica from CONACYT–Mexico. The funders had no role in study design, data collection and analysis, decision to publish, or preparation of the manuscript.

### Grant Disclosures

The following grant information was disclosed by the authors:
CONACYT–Mexico: A1-S-10669.
PAPIIT-UNAM: IN220019.
CONACYT–Mexico Ph.D: 369794.
Fondo semilla-2019: CIP-PI-029-2018-2.
FCS-Universidad Panamericana given to Victor Galvez (CVU: 421958).
Programa de Doctorado en Ciencias Biomédicas, Universidad Nacional Autónoma de Mexico: 574022/403010 (CVU: 660496).
Fondo Sectorial de Investigación para la Educación Básica from CONACYT–Mexico: CB-2017-2018-A1-S-10669-M-5248.

### Competing Interests

The authors declare that they have no competing interests.

## Author Contributions

- Gabriel Ramirez-Garcia conceived and designed the experiments, performed the experiments, analyzed the data, prepared figures and/or tables, authored or reviewed drafts of the paper, and approved the final draft.
- Victor Galvez conceived and designed the experiments, performed the experiments, analyzed the data, authored or reviewed drafts of the paper, and approved the final draft.
- Rosalinda Diaz performed the experiments, authored or reviewed drafts of the paper, contacted all participants and recruited the sample, and approved the final draft.
- Aurelio Campos-Romo conceived and designed the experiments, authored or reviewed drafts of the paper, and approved the final draft.
- Juan Fernandez-Ruiz conceived and designed the experiments, analyzed the data, authored or reviewed drafts of the paper, and approved the final draft.

## Human Ethics

The following information was supplied relating to ethical approvals (*i.e.*, approving body and any reference numbers):

The Instituto Nacional de Neurología y Neurocirugía "Manuel Velasco Suarez" and Universidad Nacional Autónoma de Mexico granted Ethical approval to carry out the study within its facilities (Ethical Application Ref: N° DIC/419/14 and N° 41/14, and N° 090/2015, respectively).

## Data Availability

The data is available at FigShare:

Ramirez Garcia, Gabriel (2021): CTRL_Avg-thickness-MocaT-Cor_mc-z.abs.th13.sig. cluster. figshare. Figure. https://doi.org/10.6084/m9.figshare.15161562.v1

Ramirez Garcia, Gabriel (2021): HD_Avg-thickness-MocaT-Cor_mc-z.abs.th13.sig. cluster. figshare. Figure. https://doi.org/10.6084/m9.figshare.15161199.v1

Ramirez Garcia, Gabriel (2021): Data_MoCA_HD_study.xlsx. figshare. Dataset. https://doi.org/10.6084/m9.figshare.15161190.v1

## Supplemental Information

Supplemental information for this article can be found online at http://dx.doi.org/10.7717/peerj.12917#supplemental-information.

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
