# Peer review of "Montreal Cognitive Assessment (MoCA) performance in Huntington’s disease patients correlates with cortical and caudate atrophy"

_PeerJ, doi:10.7717/peerj.12917_

## Round 0.1 · original submission · Minor Revisions

I agree with the reviewers’ comments. Please address their concerns in your revision.

However, I would not require omitting effect size, and I’d write p<.001 or p=.000 as indicated, not p<.000.

Note, “please revise this article...” means “please review this article.”

·

Basic reporting

no comments

Experimental design

no comments

Validity of the findings

no comments

Additional comments

Introduction
1. I suggest to the authors to revise and or update the disease stage of the patients included in the study. Please revise this article about diagnostic categories in HD. [Ross CA, Reilmann R, Cardoso F, McCusker EA, Testa CM, Stout JC, Leavitt BR, Pei Z, Landwehrmeyer B, Martinez A, Levey J. Movement disorder society task force viewpoint: Huntington's disease diagnostic categories. Movement disorders clinical practice. 2019 Sep;6(7):541.]

Statistical analysis

2. Wat was the purpose of using the Shapiro-Wilk test with standardized residuals of a linear regression analysis. Why not using the test with the variables?
3. After the linear regression, what is the decision rule for one statistical analysis or another? Please clarify as this does not seem to be relevant for the statistical analysis presented by the authors.
4. What is the purpose of using the effect size? It does not seem to have a purpose in the description of the results and in the discussion section

Results
5. For p-values, I suggest only 3 numbers after the dot (e.g., line 197). p<0.000 instead of p<0.0008
6. In the sections reporting the correlations between the MoCA and other variables, I suggest including the correlation coefficient with the p-value (e.g., rs=0.99; p=0.000)
7. In Table 1A, I suggest including the sample size in the second row of the table. Also, it is not necessary to include t-value, W-value and the effect size
8. If the significance level is defined at the beginning, it is not necessary to include symbols to indicate whether the result is significant. If the p-value is 0.009, it is clear that this value is lower than 0.05
9. In Table 1B, please indicate p-values as <0.000 and not as exponential values

Discussion

10. In the interpretation of the results, please be aware that the strength of the association is given by the correlation coefficient and not by the p-value. For example, the following result (e.g., rs=0.32; p=0.000) means a significant low correlation. The p value means that the correlation coefficient is significantly different from zero. However, the strength of the correlation is weak.

Conclusion
1. Lines 321 and 322, referring to the MoCA test as an intervention tool may be a mistake as it is a screening or assessment tool

·

Basic reporting

This is a well-written article presenting novel data. There are some details lacking in the manuscript which would help to clarify the results and conclusions

Experimental design

The study is well designed. The main drawback is the small sample size but this is acknowledged by the authors in their limitations section. Some more details would be helpful:

1. The cohort could be better characterised. Have all the HD gene carriers been given a clinical diagnosis of HD i.e. according to the Diagnostic Confidence Score? Or are they all classified as early HD just on the basis of TMS and TFC?
2. It would be useful to include range in the tables in addition to mean and SD. For example, the mean CAG is relatively high at 44. Are there any very high CAG carriers driving this mean as can happen with a small sample? It is possible that they may be atypical in their presentation as there is evidence that CAG length can influence disease trajectory
3. Which version of Freesurfer was used? Was any quality control performed on the data and if so was Freesurfer rerun on any poor quality segmentations or were there any rejections?

Validity of the findings

Do the authors have any suggestions for why there is a significant difference in intracranial volume between gene carriers and controls? Although evidence has been presented for reduced ICV in juvenile HD to my knowledge this has not been shown in adults. Importantly, software packages including Freesurfer can introduce bias in intracranial volume measurements https://pubmed.ncbi.nlm.nih.gov/23827332/

Additional comments

The authors refer to 'volume loss' but since this is just a cross-sectional study and they can't assess an ongoing atrophic process it would be more appropriate to refer to 'volume differences' between HD and controls

In the limitations section the authors suggest extending the study to more severely affected HD patients. Scanning is rarely practical beyond stage 2 due to movement artifact so I don’t know whether this is a reasonable study design.

The authors state that it was not possible to study the effect of medication on cognitive performance which is completely understandable given the small sample and variable treatments and dosing regimes etc, but many studies at least report medication usage in their cohort even if not used for analysis. This could be added to Supplementary Data, even just under broad classes such as antidepressants, antipsychotics etc. As the authors point out this is likely to have an impact on cognitive performance so it would be useful to understand roughly what proportion of the sample were receiving treatments.

---

## Round 0.2 · accepted · Accept

Thanks for your careful attention to the reviewers' comments.